# Trends in the Approval and Quality Management of Artificial Intelligence Medical Devices in the Republic of Korea

**DOI:** 10.3390/diagnostics12020355

**Published:** 2022-01-30

**Authors:** Kyoungtaek Lim, Tae-Young Heo, Jaesuk Yun

**Affiliations:** 1Ministry of Food and Drug Safety, 187 Osongsaengmyeong 2-ro, Osong-eup, Heungdeok-gu, Cheongju-si 28159, Chungbuk, Korea; indes12@gmail.com; 2College of Pharmacy and Medical Research Center, Chungbuk National University, 194-31 Osongsaengmyeong 1-ro, Osong-eup, Heungdeok-gu, Cheongju-si 28160, Chungbuk, Korea; 3Department of Information and Statistics, Chungbuk National University, Chungdae-ro 1, Seowon-gu, Cheongju-si 28644, Chungbuk, Korea; theo@cbnu.ac.kr

**Keywords:** artificial intelligence, medical device, AIMD, SaMD, diagnostic, GMP

## Abstract

Artificial intelligence (AI) is being implemented in many areas of medicine, such as patient-customized diagnosis. Growth in the artificial intelligence medical device (AIMD) field is expected in the coming years. Major countries are currently establishing systems and policies to gain a leading position in the medical artificial intelligence market. The Republic of Korea has initiated the Act on Nurturing the Medical Devices Industry and Supporting Innovative Medical Devices for the development of AIMDs and is implementing it preemptively. As a result, the country has achieved an effective strategy for coping with the COVID-19 pandemic, an increase in the number of AIMD approvals (85 approved as of September 2021), and the creation of a document pertaining to internationally harmonized guidelines on AIMD-related terms and definitions. However, in order to develop and activate more AIMD products, it is necessary to improve post-market management such as product change and quality control in addition to approval. Here, we review the current regulatory status of AIMD in the Republic of Korea and what needs to be improved for AIMD to be more developed and activated.

## 1. Introduction

With the rise of artificial intelligence (AI) and big data, and their implementation in the medical field, unprecedented medical devices are appearing in the market. A few representative examples of these devices are software as a medical device (SaMD), in which intangible software functions as a medical device; the medical big data common data model (CDM), with which hospital medical data are standardized; and digital therapeutics (DTx) that can replace existing drugs. Overall, the healthcare paradigm is shifting from standard treatments to disease prevention and precision medicine.

AI is a technology that simulates human intelligence processes, such as cognition and learning, using computer-based machine learning algorithms. The functions of the first generation of medical devices were limited to recording and displaying the results of measurements; however, AI-based medical devices have advanced and are currently used for diagnosing and determining the risk of diseases. AI-based medical devices are termed artificial intelligence medical devices (AIMDs) in the Republic of Korea, artificial intelligence and machine learning (AI/ML)-based SaMD in the United States, and machine learning-enabled medical devices (MLMDs) in the International Medical Device Regulators Forum (IMDRF) [1,2,3].

The global market for medical devices is expected to grow at a compound annual growth rate of 5.0%, from USD 408.912 billion in 2019 to USD 546.267 billion in 2025 [4]. The global market for AI healthcare is expected to grow more rapidly, from USD 2.5436 billion in 2019 to USD 11.5083 billion in 2027 at a compound annual growth rate of 49.8% [5]. If this trend continues, AI is expected to lead to substantial growth in the medical device sector. Major countries, such as the United States and European Union countries, are in a race to lead the rapidly growing medical AI market by establishing national strategies and systems. In particular, the Republic of Korea is evaluated as an exemplary case of successfully establishing a regulatory framework for AIMD in a short period of time [6]. We review the system establishment, operation and approval for the Republic of Korea to preemptively respond to AIMD and examine areas for improvement.

## 2. Trends in the Regulation of AIMDs in the Republic of Korea

### 2.1. Definition of an AIMD

To regulate AI-based products in the medical field, we first need to determine whether a product fits the definition of a medical device. If the product is identified as a medical device, it must be subjected to regulatory institutions, such as safety and performance reviews and approvals. Software that is identified as a medical device can be divided into two categories: a single software medical device called “SaMD”, and embedded software within hardware called “software in a medical device (SiMD)” [7]. AIMDs are SaMDs linked to big data and are currently being used for diagnosis.

The IMDRF, an international council organized to accelerate international regulatory harmonization related to medical devices, defines SaMDs as “software intended to be used for one or more medical purposes that perform these purposes without being part of a hardware medical device” [8].The Ministry of Food and Drug Safety (MFDS) defines an SaMD as a device which “functions in itself, conforming to the purpose of the use of the medical device and is operated in an environment equal to general-purpose computers” [9]. The MFDS comprehensively determines whether software products, including AI-based ones, are classifiable as medical devices by considering the intended use as well as the risk [1].

### 2.2. Types of AIMDs

Most AIMDs are being developed using machine learning algorithms linked to big data. Machine learning is a method whereby the software itself learns medical big data to derive the features of a disease. The medical staff can input the patients’ medical information into the medical device that applies big data and AI technology and the outputs’ auxiliary diagnosis results using features of the disease are derived using the software. Some medical software has been used to analyze medical images and to detect some features of interest; however, AIMDs learn from medical big data, reflect the results in a diagnostic algorithm, and store the data on a server in a medical institution or a cloud server.

Medical devices with AI and big data technology can be classified into AIMDs based on the clinical decision support system (CDSS), which assists the diagnostic decisions of medical professionals by analyzing medical images, and AIMDs based on the patient decision support system (PDSS) [10]. Currently, most of the AIMDs developed or licensed in the Republic of Korea are CDSS, and CDSS will be discussed here. Presently, most AIMDs are diagnostic software; however, we expect an increase in the development of more diverse products such as DTx, which replaces drugs in the treatment of diseases.

### 2.3. Systems Related to AIMDs

As mentioned above, several developed countries are preparing systems, guidelines, and policies to respond to the medical AI market. In December 2016, the United States enacted the 21st Century Cures Act, which greatly simplifies the medical device approval process. In July 2017, the FDA announced the Digital Health Innovation Plan for an efficient approach to regulation of digital technologies, including AIMD. As part of this plan, the Software Pre-Cert Pilot Program has been introduced and operated. Considering that it is difficult to commercialize low-risk SaMD, including AIMD, when applied to existing medical device regulations, this program allows companies with appropriate qualifications to voluntarily skip the medical device approval process or conduct a streamlined review. This made it possible to launch the product on the market through [10,11,12]. The European Union (EU) has previously introduced Directive 90/385/EEC on active implantable medical devices (AIMDD), Directive 93/42/EEC medical devices (MDD), and Directive 98/79/EC in vitro diagnostic medical devices (IVDD). In April 2017, the medical device regulations were divided into medical devices and in vitro diagnostic medical devices and regulated by law. The Regulation (EU) 2017/745 medical devices (MDR) came into effect in May 2021. The Regulation (EU) 2017-746/746 in vitro diagnostic medical devices (IVDR) is scheduled to be implemented in May 2022. In accordance with this regulation, AIMD applies the medical device classification system in the same way as general medical devices. A guideline was prepared to determine whether SaMD, including AIMD, is a medical device. In addition, the General Data Protection Regulation (GDPR) was revised to prepare standards for companies to use health and medical data [10,11,13,14].

In the Republic of Korea, medical devices with marked improvement in safety and effectiveness over existing ones have been designated as “innovative medical devices.” In addition, there are established and implemented laws and organizations to support the productization of medical devices.

The Act on Nurturing the Medical Devices Industry and Supporting Innovative Medical Devices (hereinafter referred to as the “Medical Device Industry Act”) was enacted on 1 May 2020. A dedicated innovative diagnostic medical device policy division, which functions as a supervisory institution, was created by the MFDS in February 2021. Under this law, the MFDS and the Ministry of Health and Welfare designate medical devices with marked improvement in safety and effectiveness over existing ones through the application of innovative medical device technology. The designated innovative medical device then undergoes a thorough review, such as the priority review, at different stages, including the development, approval, and support stages [15]. The Medical Device Industry Act allows the MFDS to certify businesses that manufacture innovative medical device software, apart from the innovative medical devices themselves, using as a reference the Pre-Certification Pilot Program operated by the U.S. Food and Drug Administration (FDA).

To support the productization of in vitro diagnostic medical devices, an approval and management system was constructed. In addition, the Act on In Vitro Diagnostic (IVD) Medical Devices was enforced on 1 May 2020 to advance the medical device system through international coordination across advanced foreign countries such as the United States and EU countries. Through this act, the scope of IVD medical devices, which had been previously restricted to IVD reagents and devices, was expanded to include software, thereby ensuring a foundation for systematic management. AI-based IVD software is currently managed according to this Act. Based on such laws and systems, the Republic of Korea was able to effectively cope with the COVID-19 pandemic using AI-based in vitro diagnostic technology [16].

The MFDS subdivides software medical device items according to the following standards [5]. First, the items are classified into embedded or independent types according to their morphological characteristics. Then, according to their functional characteristics, items are subdivided into four types: (1) control, (2) measurement, analysis, and diagnosis, (3) data conversion, transmission, and reception, and (4) display. Through this classification, 101 SaMDs have been newly installed, including in vitro diagnostic software, with a plan to expand the range of items further [17]. This classification helps predict the category under which a product that is under development would fall, thereby reducing trial and error.

### 2.4. Approval Status and Applied Technology for AIMDs

AIMDs are being developed to assist disease diagnosis based on medical images such as radiographic images, magnetic resonance imaging (MRI), computed tomography (CT), and fundus images. In particular, domestic companies are cooperating with hospitals to develop AIMDs with a focus on analyzing medical images.

Sixteen products were designated as innovative medical devices according to the Medical Device Industry Act as of October 2021 (Table 1) [18,19]. Of these, ten products are diagnostic AIMDs, and some of these products are currently being reviewed for approval.

Four manufacturing companies of innovative medical device software were certified in the Republic of Korea as of October 2021 (Table 2) [20]. All of these companies have received certifications as manufacturing companies for innovative medical device software based on AIMDs.

The approval status of AIMDs in the Republic of Korea is shown in Table 3 [19,21].

To process image data, most approved AIMDs employ convolutional neural network (CNN)-based deep learning algorithms that model the human visual center. To extract the desired features from a given image, the CNN repeats the convolution and pooling operations to produce a very large number of small images with individual features and maps the extracted feature into a fully connected layer on which the final classification is based [22].

The convolution layer produces newly transformed images by applying image filters to the original image. The pooling layer reduces the size of the images produced through the convolution layer. The fully connected layer categorizes images through convolution and pooling processes.

AIMDs analyze medical images using the following three methods: classification, detection, and segmentation (Table 4) [23].

### 2.5. Consideration of Approval for AIMDs

In the Republic of Korea, the number of approvals for AIMDs has been increasing annually since the first approval was issued in 2018. Most of these approvals have been identified as products used in diagnosis. Notably, approved products have been manufactured by domestic companies. Although it is difficult for domestic manufacturers to compete against the foreign medical device companies that dominate the Republic of Korea, there is a robust domestic AIMDs manufacturing superiority, which is largely due to the preemptive regulatory response and support for AI technology, as well as the product development efforts by the industry.

The forms of support implemented by Korea’s regulatory authorities are summarized as follows [15,24,25].

#### 2.5.1. System Preparation

As discussed above, laws such as the Medical Device Industry Act and dedicated organizations were established, and software medical device item categories were created.

#### 2.5.2. Active Utilization of Outside Experts

To review and approve products that apply the latest technologies, experts must be consulted. For prompt product approval, a council that includes MFDS and civilian experts has been in operation to discuss issues related to approval, such as document and method submissions. This has enabled the MFDS to establish the required documents and methods to certify companies manufacturing innovative medical device software.

#### 2.5.3. Publication of Industry Guidelines

Guidelines on issues such as significant changes in an AIMD have been prepared and provided to businesses or petitioners. In particular, review guidelines by stage were provided for items designated as innovative medical devices that preemptively responded to the applied technologies. Owing to these measures, the industry can preemptively identify expected problems, thereby minimizing trial and error in the product development stage and leading to prompt product approval.

#### 2.5.4. Development of Evaluation Technology for AIMDs

The MFDS has developed evaluation methods and safety criteria, as well as performance and evaluation guidelines for AIMD clinical trial plans. These guidelines were also disclosed to the industry to ensure transparency in the approval and review tasks. In 2017, the Guidelines for Review and Approval for big data and AIMDs were published for the first time. In 2020, MFDS cooperated with experts for the successful commercialization of “Doctor Answer”, a Korean-style AIMD, and developed evaluation methods and criteria for testing the safety and performance of AIMDs in the detection and diagnosis of colorectal and prostate cancer.

### 2.6. Preparation of International Standards for AIMDs

The management of AIMDs has emerged as an important topic not only in the Republic of Korea but also worldwide. International coordination on the application scope of the regulations of the relevant technology, such as the classification as a medical device, and regulatory terminology, is greatly needed. Therefore, the Republic of Korea has proposed the establishment of a working group within the IMDRF, called the Artificial Intelligence Medical Device Working Group (AIMD WG). The IMDRF is an international council organized for international regulatory coordination on medical devices and includes countries such as the Republic of Korea, the United States, and EU countries. Guidelines published by the IMDRF have been adopted by many countries [26]. In 2020, the Republic of Korea was appointed as the first chair of the AIMD WG, leading to regulatory harmonization for AIMDs. Based on its experience with the approval of AIMDs, guidelines, and development of evaluation technology, the Republic of Korea collaborated with IMDRF member countries to prepare common international guidelines for AI, titled Machine Learning-enabled Medical Devices—A Subset of Artificial Intelligence-enabled Medical Devices: Key Terms and Definitions, which is scheduled to be published in 2022 [3]. The guidelines include the scope of AI technology utilized in medical devices, the definition of representative terms, and general AI concepts. Owing to its active participation in the preparation of international standards for AIMDs, the Republic of Korea is leading the fielding of AIMDs.

### 2.7. Quality Management and Post-Market Management of AIMDs

The manufacturing of medical devices such as CT and MRI requires facilities such as worksites, test rooms, and warehouses, and manufacturing and quality management systems. Conversely, the manufacturing of AIMDs only requires programming, hence locational facilities are unnecessary. The MFDS has been revising statutes to exempt the software medical devices manufacturing industry from facility requirements. Software medical devices are expected to be manufactured freely without locational restrictions in the future.

Traditional medical devices are constantly discontinued as newer models are released; however, AIMDs can be continuously updated since they are software based. Therefore, quality management is required with each update, including a review of design, performance, and risk, otherwise, poor diagnoses may result due to product malfunctions or other problems. To prevent such problems, the development and maintenance of software medical devices should be conducted according to the manufacturing and quality management system (also called good manufacturing practice) established by the manufacturers of the medical device. In other words, software medical devices, such as general medical devices, should be controlled in their design, development, risk management, manufacturing, and corrective and preventive actions according to the manufacturing and quality management system.

In addition, security and privacy risks must be managed. Since many AIMDs employ communication technology, there is a high risk of cyber-attacks which can cause malfunctions or a leak of patient information. To address this issue, the MFDS has published application methods and a case compendium related to cybersecurity and guidelines for approval.

Furthermore, although medical devices must be labeled to provide information on the purpose of use, intangible software cannot be labeled in the same manner. Therefore, software devices are required to display a label on the screen during product operation or to provide labeling on the Internet.

For post-market management, measures are required to respond properly in the event of problems with a product. When problems in medical devices arise, actions such as suspension, recall, and disposal are taken. Most approved AIMDs are used for assisting medical personnel in disease diagnosis. Since AIMDs can be tracked, if problems occur with the AIMDs in use, then the device can be tracked and identified to implement proper measures.

## 3. Discussion

AI development is progressing at an astonishing rate, and innovative AI-based applications and services are expected to increase even faster in the future, creating value in the healthcare industry. To preemptively respond to such trends, the Republic of Korea has been preparing systems and providing support to help the rise of the AIMD industry. Comparing the United States, European Union, and the Republic of Korea, which have established systems and guidelines to respond to AIMD, the United States is the first to prepare regulations for rapid approval of AIMD and is leading the way. Rather than supporting AIMD, the European Union focused on making relevant guidelines into regulations, guidelines on whether SaMD, including AIMD, are eligible for medical devices, and standards for using health care data in product development. Although the Republic of Korea started later than these countries, it quickly prepared a system to support AIMD by referring to foreign systems, and prepared guidelines that the industry could use. Owing to these achievements, the Republic of Korea was able to effectively respond to the COVID-19 pandemic using AI-based in vitro diagnostic technology. In addition, the number of approvals issued for AIMDs has been increasing annually, and the Republic of Korea is leading efforts to establish international standards through the enactment of international guidelines.

However, despite such efforts, the regulation of medical devices still needs to be improved.

First, a regulatory framework specializing in software must be constructed. Current regulations on medical devices focus on conventional hardware and are difficult to apply to AIMDs. Updates occur frequently in AIMDs; the approval requirements of such changes after each update may become an obstacle for product development and use. Thus, we need a paradigm shift in AIMDs regulations. For example, for small modifications, we should establish measures to allow the prompt use of products by changing the requirements for follow-up reporting or reviews conducted at the time of the quality management system audit, instead of the cumbersome process of approval.

Second, we need to implement a quality management system for AIMDs. Although it was stated above that AIMDs should be operated according to the quality management system established by the manufacturer, the international standard for the quality management system of medical devices known as ISO 13485:2016 declaratively asserts points that need to be considered by the manufacturer, which is difficult to apply to an actual quality management system. More specifically, the boundary between design/development and actual production is ambiguous in the case of intangible AIMDs, and they are difficult to classify by lots or batches. While general medical devices require hygiene and insect repellent systems in the work environment, AIMDs require management of the virtual work environment, such as the operating system and network. Moreover, general medical devices are released after verifying product performance through testing and inspection; therefore, the performance of AIMDs should be verified by other methods prior to distribution, such as source code logic inspection and black or white box tests. Considering that frequent changes are characteristic of AIMDs, configuration management is required. In addition, traceability should be secured for the source code, the modification history, the persons in charge, the dataset used in learning, and the baseline (release version including subprogram and hardware), among others factors.

Concerning software development kits (SDKs), the supplier company, version, and patch history require management. The dataset used for the development of an AIMD includes the personal and sensitive information of patients, such as images and medical histories; thus, the storage method, classification of tagged information, and historical information according to dataset additions need to be managed. Strengthening cybersecurity for AIMDs should also be considered to minimize the risk of hacking attempts. AIMDs are often developed not only by existing medical device manufacturers, but also by new entities, especially startups. If these small companies establish quality management system guidelines for AIMD manufacturing facilities, addressing problems detected by a series of quality control processes during the development of AIMDs, then they would be able to easily apply ISO 13485:2016 requirements to ensure quality control.

Third, procedures should be established that allow companies to take appropriate measures when product problems occur. Although most AIMDs are currently used to assist disease diagnosis, many products accessible to the general public are expected to emerge. When problems arise with such products, it would be difficult to take measures using the current methods; therefore, separate methods of implementing response measures are needed. In particular, serious problems may occur in products with an expired service life, due to malfunctions or hacking, thereby requiring proper response measures. For example, measures should be established to require a manufacturer seeking to discontinue a product to announce the discontinuation to the users and to provide services such as updates for a given period after discontinuation. Thus, by resolving such issues and effectively managing the field of AIMDs, the development of AIMDs will increase even more actively.

## 4. Future Perspective

When the sustainability of a country’s health and medical system is threatened by the rapid increase in medical expenses due to an aging population and the increased prevalence of chronic diseases, AIMDs will become a realistic alternative that reduce the social and economic costs of healthcare and promote healthier lifestyles, irrespective of region, information access, and income.

Several AIMD technologies are being applied in the early diagnosis of retinal diseases, including macular degeneration [27,28], and the development and implementation of AIMDs are expected to increase in the future. The Republic of Korea operates a single medical insurance system, conducts health checkups for all citizens every two years, and systematically manages related data. Therefore, the accumulated big data of these health checkups can be potentially used to develop more advanced AIMDs for the early diagnosis of chronic diseases. Additionally, if health insurance is applied to AIMDs in the future, more diverse products are expected to be developed. In this review, we focused on currently developed or licensed AIMDs. Most of these products analyze images or assist in disease diagnosis, and further research on more diverse forms of AIMD is likely to be needed in the future.

## Figures and Tables

**Table 1 diagnostics-12-00355-t001:** Designation status of innovative medical devices (as of 31 October 2021).

No.	Business Name (Designation Date)	Product Name	Product Overview	AI Application Technology
1	Vuno Co., Ltd. (22 July 2020)	Ophthalmic image, computer-aided detection/diagnosis software	Software for the diagnosis of abnormal findings identified by fundus image using AI technology. Twelve abnormal findings, such as hemorrhage, macular hole glaucomatous disc change and vascular abnormality are analyzable	Developed under a CNN-based deep learning algorithm
2	Dawonmedax Co., Ltd. (22 July 2020)	Radionuclide source, therapeutic, neutron activation	A device that can selectively destroy cancer cells with boron neutron irradiation	AI not applied
3	Heuron Co., Ltd. (28 July 2020)	Neural image, computer-aided detection/diagnosis software	Software to aid in the diagnosis of Parkinson disease using brain MRI images through the application of AI technology	Developed under a CNN-based deep learning algorithm
4	Skia Co., Ltd. (3 August 2020)	Stereotaxic unit, navigation	Equipment for assisting surgeries by showing the position of breast tumor lesions through the projection of mammography CT images in 3D using augmented reality	AI not applied
5	Lunit Co., Ltd. (18 September 2020)	Medical image, computer-aided detection/diagnosis software, class 2	Software for the diagnosis of abnormal findings in chest X-ray images through the application of AI technology. Nine abnormal findings such as atelectasis, calcification, and fibrosis can be identified	Developed under a CNN-based deep learning algorithm
6	Vuno Co., Ltd. (22 September 2020)	Vital sign, analysis software	Software for the prediction of cardiac arrest through the analysis of vital signs using of AI technology	Developed under an R-CNN-based deep learning algorithm
7	Coreline Soft Co., Ltd. (17 November 2020)	Neural image, computer-aided detection/diagnosis software	Software for the diagnosis of cerebral hemorrhage that uses AI technology to analyze brain CT images	Currently undergoing review for approval
8	Medi-Whale Co., Ltd. (24 December 2020)	Cardiovascular risk assessment software	Software to analyze the risk of cardiovascular diseases based on the analysis of fundus images using AI technology	Currently undergoing review for approval
9	Medical AI (17 March 2021)	Electrocardiograph, analysis software	Software for the prediction of electrocardiogram results and cardiac arrest within 24 h using AI technology to analyze electrocardiograms	Currently undergoing review for approval
10	Meere Company, Inc. (4 May 2021)	Robotic surgical system, navigation	Automated surgery robot system used for endoscopy (laparoscopy) including cholecystectomy and prostatectomy	AI not applied
11	Laonpeople Co., Ltd. (12 May 2021)	Medical image, computer-aided detection/diagnosis software, class 2	Software for the diagnosis and assistance of sleep apnea using AI technology to analyze patient CT images and biometric information	Currently undergoing review for approval
12	TE Bios Co., Ltd. (8 July 2021)	Corneal prosthesis	Full-thickness implantable artificial cornea for patients with vision loss caused by cornea damage due to congenital or acquired factors	AI not applied Currently undergoing review for approval
13	Neurosona Co., Ltd. (27 August 2021)	Focused ultrasound stimulator system	Treatment of major depressive disorders of more than minor level by stimulating the brain with low-intensity focused ultrasound	AI not applied
14	Lunit Co., Ltd. (2 September 2021)	Breast cancer image, computer-aided detection/diagnosis software	Software for the diagnosis of breast cancer based on mammography videos by applying AI technology	Developed under a CNN-based deep learning algorithm
15	Livsmed Co., Ltd. (15 October 2021)	Surgical instruments	Composed of disposable medical suture instruments, disposable medical ligation instruments, disposable endoscopic forceps and disposable electrodes for foot control electrosurgical instruments, used as multijointed surgical instruments in surgical operations	AI not applied
16	Vuno Co., Ltd. (25 October 2021)	Electrocardiograph, analysis software	Software for the detection of heart failure and myocardial infarction, through the use of AI technology to analyze electrocardiogram test results	Currently undergoing review for approval by stage

For products undergoing review for approval, AI-applied technology, etc., are not disclosed. CNN: convolutional neural network, R-CNN: region based convolutional neural network.

**Table 2 diagnostics-12-00355-t002:** Certification status for businesses manufacturing innovative medical device software in the Republic of Korea (as of 31 October 2021).

No.	Business Name (Certification Date)	Major Products
1	Vuno Co., Ltd. (7 April 2021)	Product for the diagnosis of abnormal findings detected in fundus images using AI technology
2	Lunit Co., Ltd. (2 July 2021)	Product to assist diagnosis through the detection of abnormal findings in chest images using AI technology
3	Coreline Soft Co., Ltd. (2 July 2021)	Product to assist diagnosis using AI technology to analyze the status and amount of cerebral hemorrhage in CT images
4	Heuron Co., Ltd. (10 September 2021)	Product to assist in the diagnosis of Parkinson disease based on MRI images using AI technology

**Table 3 diagnostics-12-00355-t003:** Approval status of AIMDs in the Republic of Korea.

Classification		Total	2017	2018	2019	2020	September 2021
**Total**		85	-	4	10	50	21
Manufacturing	Subtotal	79	-	4	10	45	20
	Approval	78	-	4	10	45	19
	Notification	1	-	-	-	-	1
Import	Subtotal	6	-	-	-	5	1
	Approval	6	-	-	-	5	1
	Notification	-	-	-	-	-	-

Medical devices are divided into four classes based on their purpose of use and risk. Class 1 (low risk) is managed via notification, and classes 2, 3, and 4 are managed by approval.

**Table 4 diagnostics-12-00355-t004:** AIMD analysis methods.

Methods	Details
Classification	Assigning a medical image to one of the predefined categories
Detection	Identifying features such as a tumor in medical images. Faster regions with convolutional neural networks (R-CNN) evolved from R-CNN are generally used in detection. Faster R-CNN learns with region proposal network (RPN) and extracts a region of interest (ROI). In addition to Faster R-CNN, there also exists a technique known as YOLO (You Only Look Once), which performs detection based on regression
Segmentation	Identifying the meaningful part in a medical image

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
