# Peer review of "Trends in the Approval and Quality Management of Artificial Intelligence Medical Devices in the Republic of Korea"

_diagnostics, 2022, doi:10.3390/diagnostics12020355_

Round 1
Reviewer 1 Report
Dear Authors, please find my comments, questions, and suggestions below.
The list of references should be numbered. Please fix it.
In line 50 You provided reference [2] "Artificial Intelligence (AI) in Healthcare Market to 2027 (the insight partners)". Is this source anonymous? If not, please give a more detailed link.
You should add the list of Abbreviations at the end of the manuscript. It will be helpful to readers.
Please check the spelling, punctuation, and grammar of the manuscript. I noticed:
lines 113, 181, 194, 211, 293
Dots at the end of sentences are missing.
lines 79-80
Most AIMDs are being developed using machine learning methods linked with big data.
line 112
For this, the operation method of the “Pre-Certi” system operated...
Also, I found a typo in lines 49-50:
...the global market for AI health care is expected to grow more rapidly from 3.991 billion USD in 2019 to 107.797 billion USD in 2017 at an average annual rate of 49.8% [2].
I suppose these years need to swap.
In lines 87-90, You mentioned AIMDs based on the patient decision support system. However, the management approaches of such systems are not considered in the manuscript. You should describe it in more detail or explain why it is not considered.
You should add references to prove some of your sentences:
AI based medical device is used as the term artificial intelligence medical device (AIMD) in the Republic of Korea [*], artificial intelligence and machine learning (AI/ML) – based software as a medical device (SaMD) in the United States [*], and machine learning-enabled medical device (MLMD) in the International Medical Device Regulators Forum (IMDRF) [*].
Major countries such as the United States and the European Union are moving quickly by establishing national strategies and preparing systems to lead the rapidly growing medical AI market [*].
As mentioned above, major countries such as the United States, Canada, and EU countries are preparing systems, standards, and policies to respond to the medical AI market [*].
Could you describe here the preparing systems, standards, and policies in more detail? Also, You should discuss the management practices difference between Korea and other major countries in the Conclusion section. I think it will be interesting for readers and will allow to fully show them the level of AIMDs management in the Republic of Korea.
Author Response
[January 16, 2022]
Editor-in-Chief
Diagnostics
Dear Editor:
I wish to re-submit the manuscript titled “Trends in the Approval and Quality Management of Artificial Intelligence Medical Devices in the Republic of Korea.” The manuscript ID is diagnostics-1539859.
We thank you and the reviewers for your thoughtful suggestions and insights. The manuscript has benefited from these insightful suggestions. I look forward to working with you and the reviewers to move this manuscript closer to publication in Diagnostics.
The manuscript has been rechecked and the necessary changes have been made in accordance with the reviewers’ suggestions. The responses to all comments have been prepared and attached herewith.
We confirm that neither the manuscript nor any parts of its content are currently under consideration or published in another journal. All authors have approved the manuscript and agree with its submission to Diagnostics.
Thank you for your consideration. I look forward to hearing from you.
Sincerely,
Jaesuk Yun
College of Pharmacy and Medical Research Center
Chungbuk National University
194-31 Osongsaengmyeong 1-ro
Osong-eup, Heungdeok-gu, Cheongju-si
Chungbuk 28160
Republic of Korea
Tel.: +82-43-261-2827
Email: jyun@chungbuk.ac.kr
Response to Reviewer 1 Comments
Point 1: The list of references should be numbered. Please fix it.
Response 1: As the reviewer commented, the list of references were modified.
Please see the attachment.
Point 2: In line 50 You provided reference [2] "Artificial Intelligence (AI) in Healthcare Market to 2027 (the insight partners)". Is this source anonymous? If not, please give a more detailed link.
Response 2: Considering the copyright of the reference, it has been replaced with another reference, and the content of the text was modified.
The global market for AI healthcare is expected to grow more rapidly, from 2.5436 billion USD in 2019 to 11.5083 billion USD in 2027 at a compound annual growth rate of 49.8%.
Please see the attachment.
Point 3: You should add the list of Abbreviations at the end of the manuscript. It will be helpful to readers.
Response 3: We have added abbreviations list to the end of the manuscript.
Please see the attachment.
Point 4: Please check the spelling, punctuation, and grammar of the manuscript. I noticed:
lines 113, 181, 194, 211, 293
Dots at the end of sentences are missing.
lines 79-80
Most AIMDs are being developed using machine learning methods linked with big data.
line 112
For this, the operation method of the “Pre-Certi” system operated...
Also, I found a typo in lines 49-50:
...the global market for AI health care is expected to grow more rapidly from 3.991 billion USD in 2019 to 107.797 billion USD in 2017 at an average annual rate of 49.8% [2].
Response 4: We appreciate the suggestions. We have revised each point.
Please see the attachment.
Point 5: In lines 87-90, You mentioned AIMDs based on the patient decision support system. However, the management approaches of such systems are not considered in the manuscript. You should describe it in more detail or explain why it is not considered.
Response 5: Most AIMDs developed or licensed in Korea are CDSS. We have added the discussion about CDSS.
Please see the attachment.
Point 6: You should add references to prove some of your sentences:
AI based medical device is used as the term artificial intelligence medical device (AIMD) in the Republic of Korea [*], artificial intelligence and machine learning (AI/ML) – based software as a medical device (SaMD) in the United States [*], and machine learning-enabled medical device (MLMD) in the International Medical Device Regulators Forum (IMDRF) [*].
Response 6: As the reviewer commented, we have added the references.
AI-based medical devices are termed artificial intelligence medical devices (AIMDs) in the Republic of Korea, artificial intelligence and machine learning (AI/ML)-based SaMD in the United States, and machine learning-enabled medical devices (MLMDs) in the International Medical Device Regulators Forum (IMDRF) [1, 2, 3].
Please see the attachment.
Point 7: Major countries such as the United States and the European Union are moving quickly by establishing national strategies and preparing systems to lead the rapidly growing medical AI market [*].
As mentioned above, major countries such as the United States, Canada, and EU countries are preparing systems, standards, and policies to respond to the medical AI market [*].
Could you describe here the preparing systems, standards, and policies in more detail? Also, You should discuss the management practices difference between Korea and other major countries in the Conclusion section. I think it will be interesting for readers and will allow to fully show them the level of AIMDs management in the Republic of Korea.
Response 7: In the section “2.3 Systems Related to AIMDs”, AI and systems in the United States and European Union have been added, and in the section “3. Discussion”, a comparison of systems in the United States, European Union, and Korea has been added.
Please see the attachment.
Reviewer 2 Report
This manuscript is really a very good prepared manuscript and contains serious consideration as potential publication. I think the work will contribute to the existing literature and should be published after a minor revision:
- Abstract: The Abstract must be explained with more details
- Keywords: Those words included in the title should be removed from this section.
- Introduction: Introduction: No attractive introduction that sets a proper background knowledge of the area is presented (what has been already done? which is the real novelty of this study? which is the industrial interest of the work? no proper linking of ideas, etc). The background "why do you do this work" is not clear and should be sufficiently highlighted. you can refer to the follow related articles.
- Discussion section: few related papers were cited and discussed. The discussion should be enhanced to highlight the innovation and the impractical value of current work. The literature review of the manuscript must be improved with most updated papers (i.e. 2020 and even 2021).
- The list of references does not match the format of the journal
Author Response
[January 16, 2022]
Editor-in-Chief
Diagnostics
Dear Editor:
I wish to re-submit the manuscript titled “Trends in the Approval and Quality Management of Artificial Intelligence Medical Devices in the Republic of Korea.” The manuscript ID is diagnostics-1539859.
We thank you and the reviewers for your thoughtful suggestions and insights. The manuscript has benefited from these insightful suggestions. I look forward to working with you and the reviewers to move this manuscript closer to publication in Diagnostics.
The manuscript has been rechecked and the necessary changes have been made in accordance with the reviewers’ suggestions. The responses to all comments have been prepared and attached herewith.
We confirm that neither the manuscript nor any parts of its content are currently under consideration or published in another journal. All authors have approved the manuscript and agree with its submission to Diagnostics.
Thank you for your consideration. I look forward to hearing from you.
Sincerely,
Jaesuk Yun
College of Pharmacy and Medical Research Center
Chungbuk National University
194-31 Osongsaengmyeong 1-ro
Osong-eup, Heungdeok-gu, Cheongju-si
Chungbuk 28160
Republic of Korea
Tel.: +82-43-261-2827
Email: jyun@chungbuk.ac.kr
Response to Reviewer 2 Comments
Point 1: Abstract: The Abstract must be explained with more details
Response 1: As the reviewer commented, we have modified the Abstract.
Please see the attachment.
Point 2: Keywords: Those words included in the title should be removed from this section.
Response 2: We have erased the word in the title (Quality management system) and added diagnostic.
Please see the attachment.
Point 3: Introduction: No attractive introduction that sets a proper background knowledge of the area is presented (what has been already done? which is the real novelty of this study? which is the industrial interest of the work? no proper linking of ideas, etc). The background "why do you do this work" is not clear and should be sufficiently highlighted. you can refer to the follow related articles.
Response 3: We have added background on this study in the “Introduction section”.
Please see the attachment.
Point 4: Discussion section: few related papers were cited and discussed. The discussion should be enhanced to highlight the innovation and the impractical value of current work. The literature review of the manuscript must be improved with most updated papers (i.e. 2020 and even 2021).
The list of references does not match the format of the journal
Response 4: In this paper, some papers such as Diagnostics are cited in relation to artificial intelligence. However, there are no papers related to regulations such as approval and quality control of medical diagnosis products using artificial intelligence in Korea, which is the subject. Instead of the papers, we have provided guidelines and press releases from regulatory agencies such as Korea's MFDS and FDA and medical device-related organizations such as IMDRF.
Please see the attachment.
Point 5: The list of references does not match the format of the journal
Response 5: As the reviewer commented, we have revised the references.
Please see the attachment.

Round 2
Reviewer 1 Report
Dear Authors, thanks for improving the manuscript according to my comments. I have no more questions for you. I think now your article can be published in the Diagnostics journal in present form.